# Does Endometriosis Impact the Composition of Follicular Fluid in IL6 and AMH? A Case-Control Study

**DOI:** 10.3390/jcm12051829

**Published:** 2023-02-24

**Authors:** Khadija Kacem-Berjeb, Marouen Braham, Cyrine Ben Massoud, Hela Hannachi, Manel Hamdoun, Sana Chtourou, Linda Debbabi, Maha Bouyahia, Anis Fadhlaoui, Fethi Zhioua, Anis Feki, Nozha Chakroun, Olfa Bahri

**Affiliations:** 1Department of Reproductive Biology and Cytogenetics, Aziza Othmana Hospital, University of Tunis, Tunis 1008, Tunisia; 2Research Laboratory LR16SP01 “Infertility and Oncofertility”, Tunis 1008, Tunisia; 3Department of Gynecology and Obstetrics, Aziza Othmana Hospital, University of Tunis, Tunis 1008, Tunisia; 4Laboratory of Microbiology-Biochemistry, Aziza Othmana Hospital, University of Tunis, Tunis 1008, Tunisia; 5Department of Obstetrics and Gynecology, Cantonal Hospital Fribourg, 1708 Fribourg, Switzerland

**Keywords:** IL6, AMH, endometriosis, IVF

## Abstract

Objective: The aim of this study was to compare follicular liquid levels of IL6 and AMH in women with and without endometriosis and to evaluate their potential impact on ICSI outcomes. Materials and Methods: It is a prospective case-control study conducted on 25 women with proven endometriosis and 50 patients diagnosed with other causes of infertility. All these patients were candidates for ICSI cycles. Their follicular fluid was collected at the time of oocyte retrieval and used to evaluate IL-6 and AMH titers by electro-chemiluminescent immunoassay (Cobas e411-Roche). Results: The IL-6 levels in follicular fluid were higher in the endometriosis group than in the control group (152.3 vs. 19.9 pg/mL; *p* = 0.02). The median level for AMH was 2.2 ± 1.88 ng/mL with no statistical difference between the two groups (2.2 vs. 2.7 ng/mL, *p* = 0.41). No significant correlation between the follicular IL6 and AMH levels was observed. Conclusions: The oocyte quality seems to be preserved in patients with endometriosis with the adequate response to ovarian stimulation. High levels of follicular IL6 are in accordance with the inflammatory phenomenon of the disease; however, this increase has no impact on ICSI outcomes.

## 1. Background

Endometriosis is an estrogen-dependent inflammatory disease severely affecting women of reproductive age. Its frequency has been estimated at more than 10% in reproductive-age women [1,2,3]. This pathology is frequently misdiagnosed and its treatment is usually delayed due to the frequency of asymptomatic forms (20–25% of cases) and the absence of a reliable marker for diagnosis [4]. It is now well supported that the delay in endometriosis management affects female fertility; several studies reported a high prevalence of untreated endometriosis (30–50%) in infertile women [5]. It was also shown that this type of population is characterized by a low fecundity rate that drops from 15–20% (observed in normal reproductive age) to 2–10% [6].

If the role of endometriosis in infertility is confirmed, the mechanism by which the disease reduces fertility is still debated. Several theories have been proposed, including distorted pelvic anatomy, altered folliculogenesis, endocrine and ovulatory abnormalities. Some studies conducted in intra vitro fertilization (IVF) confirmed the last two theories; they revealed a poor ovarian reserve, a low oocyte and embryo quality and a poor implantation in women with advanced endometriosis [7,8].

Some authors have recently suggested that a dysregulated intrafollicular hormone environment as well as an abnormal intrafollicular cytokine profile may therefore be a cause of reduced fertility in endometriosis patients and reduced implantation rates of IVF [9]. They have reported increased levels of inflammatory cytokines such as interleukin (IL): IL-1β, IL-6, IL-8 and IL-18 in the peritoneal fluid and serum of patients with endometriosis. The levels of these cytokines are also elevated in the follicular fluids of patients undergoing conventional gonadotropin-stimulated IVF (cIVF) [7,8,10]. However, this hypothesis is still debated; Pellicer et al. found slightly increased concentrations of anti-mullerian hormone (AMH) and cytokines in the follicular fluid of patients with severe endometriosis [11]. IL6 represents for some authors a useful marker for endometriosis and AMH is considered as an important indicator of oocyte quality and follicle selection [12,13]. An altered hormonal environment was also reported in the follicular fluid of endometriosis patients with a decrease in concentrations of estradiol (E2) and AMH [14].

The aim of this study was to compare follicular liquid levels of IL6 and AMH in women with and without endometriosis and to evaluate their potential effect on ICSI outcomes.

## 2. Materials and Methods

### 2.1. Patients

This prospective case-control study, conducted during six months, included 75 women undergoing an ICSI cycle. The population was subdivided into two groups:-The case group comprised 25 patients with confirmed endometriosis defined according to the guidelines of the American Fertility Society (AFS-R). The confirmation was performed by laparoscopy; it showed endometriosis at stage III in 23 cases and at stage IV in 2 cases.-The control group included 50 patients matched with the case group according to age. For all these women, endometriosis was ruled out through their medical history (no dysmenorrhea, no dyspareunia), vaginal ultrasound (no visible endometriotic lesions or endometrioma, no signs of uterine adenoma or adhesions) and clinical examination. The suspected etiology of infertility for this group was male infertility in the partner (28 cases), tubal obstruction (5 cases), uterine infertility by myoma, synechia (1 case), PCOS classified as phenotype 3, i.e., having regular cycles (4 cases) and idiopathic infertility (12 cases).

All patients were informed about the procedures and gave signed informed consent.

### 2.2. ICSI Procedure

Before starting the ovarian stimulation procedures, all patients with endometriosis benefited from a blockade to their hypothalamic–pituitary–ovarian axis by an oral contraceptive method. This method lasted 6 to 8 weeks and aimed at reducing the inflammatory process.

Two stimulation protocols were used:-The antagonist protocol: ovarian stimulation was performed by injection of a daily dose of 150 to 300 international units (IU) of gonadotropin during the first two days of menstruation. Monitoring was performed on the sixth day by the measurement of Estradiol (E2) and LH levels and the count of the number of follicles by endovaginal pelvic ultrasound. On the same day, the antagonist (Cétrotide^®^ 0.25 mg, Merck, Rahway, NJ, USA) was given and then monitoring was continued according to the ovarian response. When at least three follicles with a diameter over 17 mm and E2 level = 250 pg/mL per mature follicle were obtained, an injection of 6500 IU of recombinant HCG (Ovitrelle 250^®^, Merck, Rahway, NJ, USA) was administered 36 h before oocyte recovery.-The agonist protocol: a daily injection of GnRH (DECAPEPTYL^®^ 0.1 mg, Ipsen, Merelbeke, Belgium) was administered during the luteal phase. The inhibition of the hypothalamic–pituitary axis was checked by E2 and LH levels, which should be less than 50 μg/mL and 4 IU/L, respectively. An injection daily of 150 to 300 IU of Gonadotropins was then given. A series of ICSI procedures such as oocyte retrieval, collection and preparation of the sperm, culture, evaluation and embryo transfer were performed as described previously by Kdous et al. [15].

### 2.3. Follicular Fluid IL-6 and AMH Assessment

The follicular fluid was aspirated from follicles on the day of oocyte retrieval. The analysis was performed in only the first aspirate obtained from the two ovaries. The cellular constituents of the follicular fluid were removed by centrifugation at 300× *g* for 20 min. The supernatant was then collected and frozen at −80 °C until analysis.

The IL-6 and AMH (serum (AMHs) and follicular (AMHf)) concentrations were determined by electro-chemiluminescent immunoassay (Elecsys IL6 and AMH Elecsys AMH Plus) on a Cobas e411 station (Roche Diagnostics, Mannheim, Germany) according to the manufacturer’s instructions. The IL-6 levels were assessed only in the follicular fluid.

The inter-assay coefficients of variation (CV) of these assays were less than 4%. The measure of the level biomarkers was performed in duplicate to avoid inter-assay variance.

### 2.4. Statistical Analysis

The statistical analysis was performed with the Stata program, version 13.0. The Student’s test was used to analyze continuous variables and the Chi-2 or Fischer’s exact test for categorical data.

The comparison between the two groups was performed by means of the standard deviation for quantitative variables and simple or relative frequencies for qualitative variables. A *p*-value under 0.05 was considered statistically significant.

## 3. Results

Table 1 shows the clinical and biological specificities of the studied population; the mean age was 35 years (30–39 years) with no statistical difference between the case and the control groups (*p* = 0.39). The AMH levels varied from 0.38 to 2.94 pg/mL with a median of 1.66 pg/mL. The median antral follicular count (AFC) was 10.5; it was comparable between the two groups (Table 1).

Moreover, the two groups were also comparable regarding the type of the protocol used and the dose of gonadotrophin administered (Table 2). The E2 level and number of selected follicles on the day of ovulation were similar: 1902.6 ± 2499.1 vs. 1670.2 ± 1080.5 (*p* = 0.2) and 5.4 ± 4.8 vs. 4.9 ± 4.8 (*p* = 0.2), respectively.

The comparison for the ICSI outcomes between the two groups revealed a significant difference in the number of obtained mature oocytes; it was lower in the endometriosis group than in the control one (4.7 ± 4.4 vs. 6.2 ± 3.8, *p* = 0.04). However, the fertilization rate, the number of embryos, the pregnancy rate, the implantation and the live birth rates were comparable (Table 3).

### Levels of IL-6 and AMH in the Follicular Fluids

The IL-6 levels varied from 4.2 to 2428 pg/mL in the follicular fluids (FF); higher median titers were observed in the endometriosis group than in the control group (152.3 vs. 19.9 pg/mL; *p* = 0.02) (Table 4). The extreme levels ranged from 7.2 to 2428 pg/mL for the endometriosis group and from 4.2 to 311.5 pg/mL for the control group. The patients with high levels (two in the endometriosis group and one in the control group) did not have any particular features compared to the others.

For AMHf, the median level was 2.2 ± 1.88 ng/mL in the whole studied population and no statistical difference was observed between the two groups (2.2 vs. 2.7 ng/mL, *p* = 0.41) (Table 4).

The correlation between IL-6 and AMH in FF was performed with the Pearson correlation test. The Pearson coefficient r was equal to 0.01 with no statistical significance (*p* = 0.3) showing no correlation between these two markers.

## 4. Discussion

Up to now, few studies have been interested in comparing IL-6 and AMH levels in women with or without endometriosis. The current one is the first report for Tunisian patients suffering from this pathology and consulting for IVF. The level of IL-6 in FF was higher in women with proven endometriosis; these results are in line with previous studies reporting altered levels of IL-6 in FF in endometriosis patients with or without ovarian stimulation [16]. Such high levels are not due to the exogenous administration of gonadotropin; they are probably the consequence of an inflammatory process related to the disease [17]. Actually, it can be assumed that increased amounts of intrafollicular cytokines such as IL-6 are produced by the granulosa cells or by the intrafollicular immune cells as the FF contains a broad spectrum of immune cells such as leucocytes and lymphocytes that might be increasingly produced or activated in endometriosis. However, as neither the concentration nor the activity of the intrafollicular immune cells has yet been analyzed in endometriosis patients, this assumption remains hypothetical. Besides, this result could lead to IL-6 being a good biomarker for endometriosis in symptomatic patients with pelvic pain or subfertility [18]. However, the effect of endometriosis on the follicle cytokine spectrum is poorly understood; it is unclear if high levels of cytokine have a negative impact on the follicular physiology. In order to clarify that, and to characterize the endocrine medium, AMHf levels were assessed; this parameter is considered as a reliable marker of oocyte quality and ovarian reserve [19,20]. As reported by some authors, AMHf levels were comparable between the two groups [13,14,17]. This is probably due to the large size of the majority of detected follicles. In fact, larger follicles, with a diameter over 15 mm, are known to secrete less AMHf than pre-antral and small antral ones [21]. Previously, these levels were approximately 1.5 ± 0.1 ng/mL and 1.8 ± 0.3 ng/mL for patients with mild endometriosis and controls, respectively [13,17]. However, discordant results were reported; a decreased AMHf level was observed in patients with endometriosis [22,23]. Falconer et al. estimated it at 14 pmol/mL in the endometriosis patients compared to 19.6 in patients with tubal infertility [14].

In this report, the absence of a correlation between the follicular levels of IL6 and AMHf supports the fact that the inflammatory process caused by endometriosis does not affect the intrafollicular endocrine environment and consequently the ovarian reserve. Therefore, AMH is still a good marker for the ovarian reserve and oocyte quality [24].

In response to the ovarian stimulation, the serum E2 levels and number of retrieved oocytes were comparable between the two groups. A significantly low number of mature micro-injected oocytes was detected in the “endometriosis” group; this result is consistent with recently published results [25]. However, the fertilization rate remains similar in both groups. Discordant results were reported previously; the fertilization rate was higher in patients with endometriosis (87.5%) than in those characterized by male infertility for partners (34.9%). However, this rate is lower than in patients with tubal obstruction (88.0%) [26]. Other studies showed a negative impact of endometriosis on the fertilization rate: Simon et al. estimated it at 43.5% in women with endometriosis and at 57.5% in those with tubal infertility (*p* < 0.05) [27]. The results are also controversial regarding the mean number of embryos obtained after ovarian stimulation; no difference was observed in this study between the two groups, while some authors previously described a decrease in the embryo quality for patients with a confirmed diagnosis of endometriosis suggesting an alteration of folliculogenesis [14,28]. However, Cohen et al. reported the same results when they assessed the embryo quality in a murine model by immunofluorescence and with confocal microscopy [29].

As for the pregnancy rate, a high proportion was observed in the first group but the difference was non-significant; this is consistent with previously published results [26,30]. The pregnancy rate was 32.39% for women with endometriosis (*n* = 297) and 31.06% in those with other causes of infertility (4301) [31]. This high pregnancy rate in the endometriosis group is probably the consequence of the oral contraceptive treatment given to the patients 6 to 8 weeks before starting the ovarian stimulation. This type of treatment has probably a suppressive effect on endometriotic lesions. Therefore, the inflammatory phenomenon might be incomplete; it is reduced with an improvement in endometrial receptivity [32]. Endometrial receptivity is compromised by high levels of IL6 over 4 pg/mL; it could therefore be impacted by the deleterious effect of the increased IL6 rate and not the embryo quality [33].

## 5. Conclusions

The present study suggests that oocyte quality is preserved in women with endometriosis with an adequate response to ovarian stimulation. The cytokine environment seems to be affected when oral contraceptive treatment is used. However, the increase in levels of IL6 seems to have no substantial impact on the follicular endocrine environment or on ICSI outcomes. These results should be confirmed by a study on a larger population taking into consideration the selection criteria of the control group and the effect of the freezing/thawing of the follicular fluid.

## Figures and Tables

**Table 1 jcm-12-01829-t001:** Clinical and biological characteristics of cases with proven endometriosis and without diagnosed endometriosis.

	Study Group (*n* = 75)	Case Group (*n* = 25)	Control Group (*n* = 50)	*p* *
Age (mean ± SD) (years)	35 ± 4.06	35 ±3.86	35 ± 4.19	NS
Tentative Rank (mean ± SD)	1.89 ± 1.19	2.2 ± 1.58	1.74 ± 0.92	NS
Infertility etiology (*n*):				
Endometriosis	25	25	0	
Male infertility	35	7	28	
Tubal Factor/PCOS ^1^	18	9	9	
Uterine infertility (myoma, synechia) Idiopathic infertility	1 12	0 0	1 12	
AMHs (mean ± SD) (ng/mL)	1.66 ± 1.28	1.65 ± 1.07	1.66 ± 1.38	NS
AFC ^2^ (mean ± SD)	10.5 ± 5.9	9.8 ± 6.2	10.1 ± 6.9	NS

* *p*: Chi2 test or Student *t*-test are used to compare case group and control group. ^1^ Polycystic ovarian syndrome. ^2^ Antral follicle count.

**Table 2 jcm-12-01829-t002:** Parameters of the ovarian stimulation in cases with proven endometriosis and without diagnosed endometriosis.

	The Study Population (*n* = 75) Mean ± SD	Case Group (*n* = 25) Mean ± SD	Control (*n* = 50) Mean ± SD	*p* *
Stimulation Protocol: - Antagonist - Agonist	81.33% 18.67%	84% 16%	80% 20%	0.74 0.91
Total dose of administered gonadotrophin	2240 ± 680	2320 ± 780	2210 ± 590	0.72
Endometrial thickness (mm)	9.50 ± 1.97	9.64 ± 2.48	9.43 ± 1.61	0.33
E2 level on the day of ovulation trigger (pg/mL)	1747.69 ± 1676.57	1902.68 ± 2499.14	1670.2 ± 1080.52	0.28
Number of follicles ≥15 mm	5.09 ± 3.85	5.4 ± 4.88	4.94 ± 3.26	0.22

* *p*: Chi2 test or Student *t*-test are used to compare case group and control group.

**Table 3 jcm-12-01829-t003:** Comparison of ICSI outcomes in cases with proven endometriosis and without diagnosed endometriosis.

	The Study Population (*n* = 75) Mean ± SD	Case Group (*n* = 25) Mean ± SD	Control (*n* = 50) Mean ± SD	*p* *
Number of oocytes retrieved	8.38 ± 5.65	8.04 ± 6.39	8.56 ± 5.31	NS
Number of mature oocytes	5.74 ± 0.07	4.72 ± 4.41	6.26 ± 3.84	0.04
Number of 2PN stage embryos	3.62 ± 3.02	3.63 ± 3.79	3.62 ± 2.66	NS
Fertilization rate	60.55 ±23.17	64.07 ± 25.91	59.92 ± 24.00	NS
Number of embryos ready for transfer on day 2 or day 3	3.47 ± 2.84	3.63 ± 3.81	3.4 ± 2.33	NS
Top embryo rate	42.8 ± 37.67	41.71 ± 40.01	37.64 ± 32.15	NS
Number of transferred embryos	1.86 ± 0.84	1.66 ± 1.08	1.94 ± 0.73	NS
Pregnancy/transfer rate	35.38%	40%	34%	NS
Implantation rate per transfer	23.33%	30.15%	20.40%	NS
Live birth rate	23.61%	28.57%	22.44%	NS

* *p*: Chi2 test or Student *t*-test are used to compare case group and control.

**Table 4 jcm-12-01829-t004:** Levels of IL6 and AMH in follicular fluid for the two groups.

Levels of Biological Parameters in Follicular Fluid	Case Group (*n* = 25)	Control Group (*n* = 50)	*p* *
Follicular IL6 level (pg/mL)	152.3 ± 485.4	19.9 ± 42.9	0.02
Follicular AMH level (ng/mL)	2.2 ± 2.3	2.7 ± 3.3	NS

* *p*: Chi2 test or Student *t*-test are used to compare the case group and the control group.

## Data Availability

No data available.

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
