# Peer review of "Does Endometriosis Impact the Composition of Follicular Fluid in IL6 and AMH? A Case-Control Study"

_jcm, 2023, doi:10.3390/jcm12051829_

Round 1

Reviewer 1 Report

The authors checked the follicular liquid levels of IL6 and AMH in women with and without endometriosis. This is an interesting topic. There are some comments that should be considered.

For the selection of endometriosis patients (case group), authors should specify the stages of the disease for their patients. The results of the study should be compromised based on the different stages of endometriosis.

In addition, the selection of the control group is not suitable for this study. Since the patients with PCOS, uterine disorders should not allow for the inclusion of this study.

According to the results and table 3, fertilization, implantation, pregnancy, and live birth rates are higher in the case group (patients with endometriosis) than the control group (although non-significant). It is very strange and does not seem correct. These results should be the result of the incorrect selection of two groups.

The sample size of the case group is lower than 30 number and half of the control group. This can induce bias in the results or reduce the power of the study.

Table 3; the "life birth rate" is not correct. The live birth rate is the correct term.

Figure 4 does not add any considerable results for this manuscript and there is no need for presenting this figure.

Authors should assess the correlation between AMH and IL-6 concentrations and ICSI outcomes in two groups.

Author Response

The authors checked the follicular liquid levels of IL6 and AMH in women with and without endometriosis. This is an interesting topic. There are some comments that should be considered.

For the selection of endometriosis patients (case group), authors should specify the stages of the disease for their patients. The results of the study should be compromised based on the different stages of endometriosis.

  • For this comment, we specified in the manuscript (section “Materials and Methods », lines 66-68) the stage of endometriosis for all patients: “Confirmation was performed by laparoscopy; it showed endometriosis at stage III in 23 cases and at stage IV in 2 cases ».
  • As all patients had advanced endometriosis and the most of them with the same stage, the obtained results seem not to be compromised and so should be considered.

In addition, the selection of the control group is not suitable for this study. Since the patients with PCOS, uterine disorders should not allow for the inclusion of this study.

  • The control group was matched to the case group by age; In fact, this is the main criterion used in different studies on this subject. For our control group, the majority were characterised by male factor infertility (N = 28) or idiopathic infertility (N = 12). The only four cases with PCOS were phenotype 3 so with regular cycles. This type of patient can be considered as part of an idiopathic cause (section “Materials and Methods », line 75). 

According to the results and table 3, fertilization, implantation, pregnancy, and live birth rates are higher in the case group (patients with endometriosis) than the control group (although non-significant). It is very strange and does not seem correct. These results should be the result of the incorrect selection of two groups.

  • For this comment, we verified all our results and confirmed them.
  • It should be noted that in the literature, various studies have reported similar results. We can cite as  reference:  Feichtinger M, and al . Endometriosis and cumulative live-birth rate after fresh and frozen IVF cycles with single embryo transfer in young women: no impact beyond reduced ovarian sensivity- a case control study. Journal of Assisted Reproduction and Genetics. 2019; 36 :1649-1656 (reference 31 in our manuscript)

The sample size of the case group is lower than 30 number and half of the control group. This can induce bias in the results or reduce the power of the study.

  • Initially our goal was to reach 30 patients, but during the study period and taking into account the inclusion criteria, we could only collect 25 patients. However, the sample size is large enough for the results to be valid.
  • In the case-control study, the statistical analysis recommends that the size of the control group should be at least twice that of the case group

Table 3; the "life birth rate" is not correct. The live birth rate is the correct term.

  • We thank the reviewer for this comment and we corrected this mistake in the manuscript.

Figure 4 does not add any considerable results for this manuscript and there is no need for presenting this figure.

  • As recommended, we deleted the figure and replaced it by the following sentence:Correlation between IL-6 and AMH in FF was performed by the Pearson correlation test. The Pearson coefficient r was equal to 0.01 with no statistical significance (p =0.3) showing no correlation between these two markers (section “Results », lines 153-156). 

Authors should assess the correlation between AMH and IL-6 concentrations and ICSI outcomes in two groups.

  • In fact, this comment is relevant. However, for the analysis to be powerful, it is recommended to use the logistic regression in order to study the correlation between these markers and the rate of live births (which is a qualitative parameter). This  requires a larger population.

Reviewer 2 Report

This paper is dealing with IVF/ICSI as treatment of infertility in women with endometriosis and hardly with AMH and IL-6 as mentioned in the title. Results of mean AMH levels are reported in Table 1 with patient characteristics and IL-6 levels are only reported as a range in the text. There are a few patients with high IL-6 levels in Figure 1, but no details on these cases are provided. The second part of the Discussion focuses on the IVF results and should have been more concentrated on AMH and IL-6.

Author Response

This paper is dealing with IVF/ICSI as treatment of infertility in women with endometriosis and hardly with AMH and IL-6 as mentioned in the title.

  • Results of mean AMH levels are reported in Table 1 with patient characteristics and IL-6 levels are only reported as a range in the text.

Serum AMH was measured as part of the initial assessment of the patients and is therefore reported in Table 1, which represents the clinico-biological parameters of the patients.

  • To take into account this comment, we added table 4 in the manuscript. It reported IL6 and AMH concentrations in follicular fluids in the two groups.

There are a few patients with high IL-6 levels in Figure 1, but no details on these cases are provided.

  • The figure 1 is deleted as recommended by the reviewer 1. However, in response to this comment, all elevated follicular IL 6 levels, were rechecked during the study and confirmed. These patients did not have any particular features compared to the others.

The second part of the Discussion focuses on the IVF results and should have been more concentrated on AMH and Il 6

  • In the Discussion section, we reported results of the majority of studies related to this subject.  However, we also added more details as recommended in lines 165-173

Round 2

Reviewer 2 Report

Table 4 has been added now, but without ranges. No information about the cases with extremely high IL6 levels
